# Causality of anthropometric markers associated with polycystic ovarian syndrome: Findings of a Mendelian randomization study

Kushan De Silva[1]*, Ryan T. Demmer[2,3], Daniel Jönsson[4,5], Aya Mousa[1], Helena Teede[1], Andrew Forbes[6], Joanne Enticott[1]

1 Monash Centre for Health Research and Implementation, School of Public Health and Preventive Medicine, Faculty of Medicine, Nursing, and Health Sciences, Monash University, Clayton, Australia, 2 Division of Epidemiology and Community Health, School of Public Health, University of Minnesota, Minneapolis, Minnesota, United States of America, 3 Department of Epidemiology, Mailman School of Public Health, Columbia University, New York, NY, United States of America, 4 Department of Clinical Sciences, Faculty of Medicine, Lund University, Malmö, Sweden, 5 Public Dental Service of Skane, Lund, Sweden, 6 Biostatistics Unit, Division of Research Methodology, School of Public Health and Preventive Medicine, Faculty of Medicine, Nursing and Health Sciences, Monash University, Melbourne, Australia

* kushan.ranakombu@monash.edu

**Data Availability Statement:** All data used in this study are freely available in the IEU GWAS database at: https://gwas.mrcieu.ac.uk/.

## Abstract

### Introduction

Using body mass index (BMI) as a proxy, previous Mendelian randomization (MR) studies found total causal effects of general obesity on polycystic ovarian syndrome (PCOS). Hitherto, total and direct causal effects of general- and central obesity on PCOS have not been comprehensively analyzed.

### Objectives

To investigate the causality of central- and general obesity on PCOS using surrogate anthropometric markers.

### Methods

Summary GWAS data of female-only, large-sample cohorts of European ancestry were retrieved for anthropometric markers of central obesity (waist circumference (WC), hip circumference (HC), waist-to-hip ratio (WHR)) and general obesity (BMI and its constituent variables–weight and height), from the IEU Open GWAS Project. As the outcome, we acquired summary data from a large-sample GWAS (118870 samples; 642 cases and 118228 controls) within the FinnGen cohort. Total causal effects were assessed via univariable two-sample Mendelian randomization (2SMR). Genetic architectures underlying causal associations were explored. Direct causal effects were analyzed by multivariable MR modelling.

### Results

Instrumental variables demonstrated no weak instrument bias (F > 10). Four anthropometric exposures, namely, weight (2.69–77.05), BMI (OR: 2.90–4.06), WC (OR: 6.22–20.27), and

**Funding:** KDS is supported by a PhD scholarship funded by the Australian Government under Research Training Program (RTP). The funders had no role in study design, data collection and analysis, decision to publish, or preparation of the manuscript.

**Competing interests:** The authors have declared that no competing interests exist.

HC (OR: 6.22–20.27) demonstrated total causal effects as per univariable 2SMR models. We uncovered shared and non-shared genetic architectures underlying causal associations. Direct causal effects of WC and HC on PCOS were revealed by two multivariable MR models containing exclusively the anthropometric markers of central obesity. Other multivariable MR models containing anthropometric markers of both central- and general obesity showed no direct causal effects on PCOS.

## Conclusions

Both and general- and central obesity yield total causal effects on PCOS. Findings also indicated potential direct causal effects of normal weight-central obesity and more complex causal mechanisms when both central- and general obesity are present. Results underscore the importance of addressing both central- and general obesity for optimizing PCOS care.

## Introduction

Polycystic ovarian syndrome (PCOS) is the most common endocrine disorder in women of reproductive age [1], with a prevalence ranging 8%-18% among this group [2]. It is also the major cause of anovulatory female infertility [3]. While a multitude of reproductive and metabolic abnormalities associate with PCOS, its convoluted etiology, presumably multifactorial and heterogeneous, is still not completely known. Classic clinico-pathologic features of PCOS comprise hyperandrogenism, oligo-anovulation, excessive weight or obesity and a range of metabolic manifestations such as glucose intolerance, insulin resistance, and dyslipidemia while a number of obstetric, cardiometabolic, oncological, and psychological complications may also ensue [4]. A recent review highlighted the correlation between obesity, hyperandrogenism, and insulin resistance, and suggested these might act in concert forming a vicious cycle to induce PCOS [5]. In addition, neuroendocrine changes such as gonadotropin secretory abnormalities [6], fetal programming alterations within the intrauterine microenvironment [7], genetic- and epigenetic factors [8], an array of environmental predictors [9] and certain inflammatory regulators [10] have also been implicated in the pathogenesis of PCOS. Constraints to the present PCOS care practices include a lack of uniform diagnostic criteria, limitations intrinsic to the evidence base emanating from observational epidemiological studies such as the concerns of confounding and reverse causality [11] as well as diagnostic delays and difficulties [12].

It should be noted that a substantial proportion of the PCOS-related evidence base stems from observational studies which are precluded by various biases and unmeasured confounding. However, causal inference is fundamental to unravelling disease etiologies and mechanistic underpinnings. The gold standard of etiological inference remains randomized controlled trials which could overcome key limitations of observational studies. Given the practical drawbacks of randomized trials especially exorbitant financial costs and longer time spans entailed, Mendelian randomization (MR) methods offer a viable and robust alternative to inferring causality using observational data. Using genetic variants as instrumental variables (IVs) and leveraging on their strengths, especially stability and random assortment of alleles, the causality between risk factors and diseases can be plausibly deduced by MR. With the increasing availability of genome-wide association studies (GWAS) data, MR is making considerable contributions to uncovering complex, polygenic disease etiologies [13]. Well-conducted MR

studies meeting the three key assumptions of relevance, independence, and exclusion restriction could thus provide high-level evidence on causality [14]. To this end, guidelines such as the Strengthening the Reporting of Observational Studies in Epidemiology Using Mendelian Randomization (STROBE-MR) [15] have been developed to facilitate uniform and comprehensive presentation of MR studies.

Obesity is frequently observed in women with PCOS, predominantly in the form of abdominal obesity, aggravating metabolic and reproductive sequalae and PCOS-associated complications [16]. Available evidence suggests that the close link between obesity and PCOS is presumably mediated by multiple mechanisms such as insulin resistance-driven metabolic changes, steroidogenic and reproductive effects of hyperinsulinemia, and augmented adipokine synthesis by subcutaneous and visceral fat [17]. De Segher et al underscored the absence of a gonadotropic and/or ovarian disorder in a majority of girls with PCOS and theorized that central obesity and central adiposity are the main drivers of PCOS development [18]. Putative genetic links between obesity and adiposity with PCOS are reinforced by current evidence [19–21], for instance, the influence of *FTO* gene variants [22, 23]. A caveat in the relationship between obesity and PCOS is that obesity alone may be neither a necessary nor a sufficient condition for its development among all PCOS-susceptible females [19], given the presence of lean PCOS phenotypes [24], exemplifying the complex, multifactorial etiology of PCOS.

Notably, multiple MR studies revealed that obesity potentially causes PCOS, all of which exclusively used body mass index (BMI) as the surrogate anthropometric measure of obesity [25–28]. However, BMI is essentially a marker of general obesity whereas central obesity in particular seems to confer a formidable influence on PCOS pathogenesis [16–18]. Moreover, the use of multiple anthropometric traits could yield not only complementary but also exclusive information. For example, a recent consensus statement highlighted the unequivocal evidence that waist circumference (WC) provides additive and independent information to BMI and recommended that WC be used as a vital sign in clinical practice [29]. Also, concordant and consistent findings emanating from MR analyses using multiple anthropometric markers of central- and general- obesity could strengthen the evidence pertaining to their putative causal roles in ensuing PCOS. Anthropometric traits are customarily exploited in MR studies as valid and reliable surrogate markers for operationalizing central- and general obesity [30–32]. In these studies, multiple anthropometric markers were incorporated in univariable MR to gain evidence on causal roles of central- and general obesity [30–32].

In this study, we examined existing data from GWAS, and aimed to investigate the causality of multiple anthropometric markers of central- and general- obesity associated with PCOS via MR analyses. Specifically, we conducted univariable two-sample MR (2SMR) to determine total causal effects, multivariable MR to assess direct causal effects, and bidirectional MR- analyses to evaluate reverse causality, using anthropometric traits of central obesity (WC, hip circumference (HC), waist-to-hip ratio (WHR)) and general obesity (BMI and its constituent variables–weight and height).

## Materials and methods

This study was conducted according to STROBE-MR guidelines [15], as described in **S1 Table**. Formal ethics approval was not required, being an analysis of publicly available, deidentified, summarized data.

### Data sources for exposures

For the six anthropometric traits selected as exposures, we performed a comprehensive search on the IEU Open GWAS Project database (https://gwas.mrcieu.ac.uk/) to find the most

suitable GWAS summary data for MR analyses. Given the female-exclusivity of PCOS and sex-based variations in genetically-proxied anthropometric traits [33], we retrieved female-only GWASs. In order to alleviate confounding by ancestry and population stratification effects, we resorted to cohorts with participants of European ancestry. We prioritized GWASs with large sample sizes and ultimately identified appropriate studies from the **G**enetic **I**nvestigation of **AN**thropometric **T**raits (GIANT) consortium (**Table 1**).

For weight, we selected the study with the GWAS-ID "ieu-a-107" reporting summary statistics from a meta-analysis of 46 cohorts over 73137 females of European ancestry and 2747007 SNPs. Data had been age-adjusted and details of the original study is available elsewhere [33].

The study selected for height with the GWAS-ID "ieu-a-97" report summary information from a meta-analysis of 46 studies spanning 73137 females of European ancestry and 2748546 SNPs. Data had been age-adjusted and further information of the study is published elsewhere [33].

For BMI, the study with the GWAS-ID "ieu-a-974" was selected, which report summary data from a large-scale meta-analysis of 82 GWASs over 171977 females of European ancestry and 2494613 SNPs. Data had been age-adjusted and details are given elsewhere [34].

For WC and HC, we selected studies with the GWAS-IDs "ieu-a-63" and "ieu-a-51" respectively, which report pooled statistics from large-scale meta-analyses of 127997 females of European ancestry and 2444355 SNPs. Data had been adjusted for age and other study-specific covariates and details are available elsewhere [35].

The study with the GWAS-ID "ieu-a-75" was selected for WHR, which report pooled results from a meta-analysis encompassing 118003 females of European ancestry and 2466102 SNPs. Data had been adjusted for age and other study-specific covariates and details are available elsewhere [35].

## Data source for PCOS

In order to avoid sample-overlapping with exposures, we selected PCOS GWAS summary statistics from a different source i.e. the FinnGen cohort on the IEU Open GWAS Project database. The FinnGen study entails a growing repository of genomic and clinical data emanating from a nationwide network of Finnish biobanks (https://finngen.gitbook.io/documentation/). We used the GWAS with the specific ID "finn-b-E4_POCS" (E4_POCS is the FinnGen pheno-code for PCOS) which consisted of 118870 samples (642 cases; 118228 controls) and 16379676 genotyped SNPs in total (**Table 1**). All PCOS cases were clinically diagnosed from hospital discharge registries and cause of death registries using female-specific clinical endpoints (ICD-10: E282, ICD-8: 25690).

**Table 1. Female-only, European ancestry GWASs retrieved from the IEU Open GWAS project to be included as exposures (anthropometric markers) and the outcome (PCOS) in two-sample Mendelian randomization analyses.**

| Exposures and outcome | GWAS ID | Consortium | Sample size | No: of SNPs | Reference |
|---|---|---|---|---|---|
| **Exposures: Anthropometric markers** | | | | | |
| Weight (kg) | ieu-a-107 | GIANT | 73137 | 2747007 | Randall et al., 2013; PMID = 23754948 |
| Height (m) | ieu-a-97 | GIANT | 73137 | 2748546 | Randall et al., 2013; PMID = 23754948 |
| Body mass index (BMI) (kg/m$^2$) | ieu-a-974 | GIANT | 171977 | 2494613 | Locke et al., 2015; PMID = 25673413 |
| Waist circumference (WC) (cm) | ieu-a-63 | GIANT | 127997 | 2444355 | Shungin et al., 2015; PMID = 25673412 |
| Hip circumference (HC) (cm) | ieu-a-51 | GIANT | 127997 | 2444355 | Shungin et al., 2015; PMID = 25673412 |
| Waist-to-hip ratio (WHR) | ieu-a-75 | GIANT | 118003 | 2466102 | Shungin et al., 2015; PMID = 25673412 |
| **Outcome** | | | | | |
| Polycystic ovarian syndrome (PCOS) | finn-b-E4_POCS | FINNGEN | 118870 (642 cases; 118228 controls) | 16379676 | N/A |

## Selection of genetic variants as IVs

The SNPs selected as IVs need to fulfil three key assumptions for MR to yield valid results: (1) strongly associated with the exposure (relevance); (2) not associated with the outcome due to confounding (independence); (3) affect the outcome only through the exposure (exclusion restriction). In order to fulfil the first criterion of relevance, we selected biologically and statistically plausible SNPs at a genome-wide significance threshold of $p < 5e\text{-}08$. We assessed the statistical power of individual SNPs and the potential weak instrument bias via F-statistic, as defined below.

$F = v^2 \times (n - 2)/(1 - v^2)$, in which $v^2$ indicates the variance of the exposure phenotype attributable to a given SNP and $n$ stands for the sample size. Variance estimates were calculated using the following formula:

$$v^2 = 2 \times \beta^2 \times eaf \times (1 - eaf)/\left[2 \times \beta^2 \times eaf \times (1 - eaf) + 2 \times se^2 \times n \times eaf \times (1 - eaf)\right],$$

in which $\beta$ denotes the per-allele effect size of the association between a given SNP and the exposure phenotype, *eaf* stands for the effect allele frequency, and *se* is the standard error of $\beta$ [36]. Weak instrument bias may ensue when $F < 10$ [37].

We minimized confounding through the inclusion of female-only cohorts of European ancestry from a single consortium while the summary data had been adjusted for age and other study specific covariates. The assumption of exclusion restriction may be violated in the presence of horizontal pleiotropy, especially unbalanced (directional) horizontal pleiotropy–where one or more IVs exert a net effect on the outcome via a pathway not involving the exposure, biasing the MR estimate. In order to overcome horizontal pleiotropy, we conducted robust MR methods, post-hoc pleiotropy analyses, heterogeneity tests and outlier analyses [38], the details of which are provided later in the manuscript.

As the absence of linkage disequilibrium (LD) is a prerequisite for most MR methods, we performed clumping to prune statistically plausible IVs ($p < 5e\text{-}08$), selecting those SNPs with LD-$R^2$ < 0.001 and clumping-distance > 10000 kb. When any exposure SNP was not present in the outcome data, we included proxy SNPs via LD tagging, with following specifications: minimum LD-$R^2$ value = 0.8; minor allele frequency threshold for aligning palindromic SNPs = 0.3. We performed allele harmonization by aligning strands for all SNPs including palindromes, to ensure that the effects of the SNPs on the exposure correspond to the same allele as their effects on the outcome.

## 2SMR analyses

We applied four 2SMR methods i.e. inverse variance weighted (IVW) method using multiplicative random effects model as the primary analysis along with three robust methods: MR-Egger (MRE), weighted median (WME), and weighted mode (WMO) [38].

The IVW method is the recommended main MR method to be used with summarized data and multiple, uncorrelated genetic variants, since it is the most efficient approach in the presence of valid IVs, yielding causal estimates that are accounted for heterogeneity [38]. This method pools Wald ratio estimates for individual SNPs using inverse of the variance as weights.

The MRE method provides an asymptotically consistent causal effect measure adjusted for horizontal pleiotropy by pooling individual SNP-specific Wald ratios via an adapted Egger regression. In fact, its regression intercept is an estimate of the net pleiotropic effect which can therefore be used to assess horizontal pleiotropy. It allows for invalid IVs, provided the Instrument Strength Independent of Direct Effect (INSIDE) assumption (i.e. instrument strengths are independent of horizontal pleiotropic effects) holds true. However, this method is

underpowered in the presence of relatively homogeneous SNP-exposure effect sizes, suscepti-
ble to regression dilution bias, and the causal estimation is heavily affected by outliers [38–40].

Using the weighted median of Wald ratios, the WME method produces an asymptotically
consistent estimate of the causal effect, provided 50% or more of the variants are valid IVs that
do not violate the exclusion restriction criterion [38]. The WMO method clusters SNPs into
groups based on the similarity of their individual ratio estimates, calculates the inverse vari-
ance weighted number of SNPs in each cluster, and produces a causal estimate based on the
cluster having the largest weighted number of SNPs [41]. Both WME and WMO methods
require some genetic variants to be valid instruments and are robust to outliers [38].

Results from 2SMR analyses were summarized in tabular format, along with odds ratios
(OR) and 95% confidence intervals (CI) for all β estimates.

## Method comparison plots

Scatter plots and trend lines pertaining to different 2SMR methods were generated for each
anthropometric exposure-outcome analysis. Slopes and directions of trend lines represent the
magnitudes and directions of causal estimates, respectively.

## Single SNP analyses

Causal effects of each SNP were determined individually and were visualized in forest plots
along with pooled estimates using all SNPs, under IVW and MRE methods. With respect to
each exposure, we investigated for the presence of any significantly causally associated individ-
ual SNPs at Bonferroni multiple testing corrected $p$-value thresholds.

## Leave-one-out sensitivity analyses

We conducted leave-one-out sensitivity analyses to assess whether causal estimates were signif-
icantly influenced by a single SNP. Wald ratios from IVW-MR analyses conducted excluding
each SNP as well as pooled IVW-MR estimates encompassing all SNPs were visualized in for-
est plots.

## Heterogeneity analyses

Heterogeneity is a measure of the consistency of the causal estimate across all SNPs whereby
lower heterogeneity is indicative of a reliable MR estimate. We assessed heterogeneity via
Cochran's Q statistic and associated $p$-values. Funnel plots were also generated to visually
assess heterogeneity. Lower values on the y-axis denote less precise estimates while those with
increasing precision tend to 'funnel' in. Larger distributions suggest heterogeneity which may
have resulted from horizontal pleiotropy whereas asymmetric distributions are indicative of
unbalanced horizontal pleiotropy.

## Analysis of horizontal pleiotropy and outliers

The MRE regression intercept is an estimator of the magnitude of horizontal pleiotropy [39].
For each anthropometric exposure, we calculated the MRE intercept, its standard error, and
directionality $p$-value. The Mendelian Randomization Pleiotropy RESidual Sum and Outlier
(MRPRESSO) method presents a unified framework to assess pleiotropy and outliers via a
three-step process: identification of pleiotropy and outliers (MRPRESSO global test); rectifica-
tion of pleiotropy by removing outliers (MRPRESSO outlier test); analysis of the distortion in
the causal estimate before and after removing outliers (MR-PRESSO distortion test) [42]. We
applied the MRPRESSO method to evaluate pleiotropy and outliers with respect to all

exposure-outcome associations. Radial plots have been proposed as an improved strategy for visualizing outliers in 2SMR analyses [43]. We generated radial plots for all exposure-outcome associations that were investigated with 2SMR in this study.

Analyses were conducted in *R* (version 4.1.2) [44], using "TwoSampleMR" [45], "MRPRESSO" [42], and "RadialMR" [43] packages.

## Exploring genetic architectures underlying causality

Subsequent to causality testing, MR studies increasingly conduct downstream analyses to explore genetic underpinnings [46, 47]. For instance, previous MR studies showed that functional annotations of highly significant SNPs that act as major drivers of causality in MR models could assist in gaining etiologic insights [46] and identifying putative drug targets [47]. For significant causal associations, we predicted nearest gene(s)/transcriptional start site(s) ascribed to corresponding SNPs in harmonized datasets, using the Open Targets Genetics Portal [48]. In order to identify shared and non-shared genetic architectures underlying these significant causal associations, we explored overlapping and non-overlapping SNPs and genes across exposures.

## Multivariable MR analyses

Standard, univariable MR estimates the total causal effect of an exposure on the outcome whereas multivariable MR measures the direct causal effect of an exposure on an outcome, conditioned on other exposure(s) [49]. Multivariable MR is an advancement of MR to account for IVs that are associated with multiple exposures. Its application is envisaged in situations such as when the exposures are biologically related and when exposures potentially formulate a network of causal effects. Specifically, its benefits in the presence of complexities such as confounders, colliders, vertical or functional pleiotropy, and mediation have been demonstrated [49, 50]. The three assumptions of IVs in multivariable MR are that the SNPs: associate with one or more exposures (relevance); do not associate with the outcome via confounding (independence); do not to affect the outcome except via its association(s) with single or multiple exposure(s) being assessed (exclusion restriction) [38].

In order to appraise direct causal effects, we performed multivariable MR analyses with various combinations of anthropometric markers that demonstrated significant total causal effects in preceding, univariable 2SMR analyses. This was performed using the multivariable inverse variance weighted method deployed through the 'mr_mvivw' function in the 'MendelianRandomization' R package [51, 52]. In brief, this method entails the following sequence of steps: acquisition of IVs for each exposure; collation of these into a set of all IVs; clumping them to alleviate LD between variants across exposures; re-extraction of all the final, clumped IVs from all of the exposures; harmonization of variants to be on the same reference allele; application of multivariable MR upon the harmonized data. Specifically, multivariable weighted linear regression was conducted and the direct causal effect was estimated by regression of the associations with PCOS on the associations with the anthropometric exposures, with the intercept set to zero and weights being the inverse variances of the associations with PCOS. A multiplicative random effects model was assumed in all multivariable analyses, accounting for heterogeneity between causal estimates and allowing for over-dispersion in regression models. Heterogeneity statistic (Cochran's Q) and associated *p*-value tested the null hypothesis that all genetic variants estimated the same causal estimate and its rejection indicates the presence of single or multiple pleiotropic variants.

## Bidirectional MR analyses

In order to assess the reverse causality hypothesis (i.e. whether PCOS in turn causes central or general obesity), we conducted a set of univariable 2SMR analyses, taking PCOS as the exposure and each anthropometric marker as the outcome. For this purpose, we used the same data sources described previously. We applied the same criteria and specifications as stated previously for the selection of genetic variants as IVs, while 2SMR analyses followed the same methodological approach detailed earlier.

# Results

## Genetic variants selected as IVs

Harmonized datasets containing details of the SNPs selected as IVs for each exposure are presented in **S2 Table**. Eleven SNPs underlying the weight-PCOS association were identified, which comprised no proxies and no palindromes. We included 51 SNPs with respect to the height-PCOS association, which consisted of no proxies and 5 palindromes. Thirty-seven SNPs were identified with respect to the BMI-PCOS association, which comprised 2 proxies and 2 palindromes. With respect to WC-PCOS and HC-PCOS associations, we identified 17 identical SNPs, which comprised no proxies and 2 palindromes. We identified 23 SNPs underlying the WHR-PCOS association, which consisted of no proxies and a single palindrome. The F-statistic of all IVs across the exposures was > 10, indicating no substantial weak instrument bias.

## 2SMR results

Results from 2SMR analyses are presented in **Table 2**. Four exposures, namely, weight, BMI, WC, and HC, were significantly causally associated with PCOS.

According to the IVW method, a unit (1 kg) increase in weight was associated with a significant increase in PCOS (OR = 2.69; 95% CI = 1.04, 6.95; $p$ = 0.0413). As per the MRE model, a unit increase in weight yielded an OR of 77.05 (95% CI = 3.14, 1890.64; $p$ = 0.0260).

According to the IVW method, a unit (1 kg/m$^2$) increase in BMI was associated with a significant increase in PCOS (OR = 4.06; 95% CI = 2.29, 7.22; $p$ = 1.68E-06). As per the WME model, a unit increase in BMI yielded an OR of 2.90 (95% CI = 1.22, 6.90; $p$ = 0.0162).

The IVW method revealed that WC was significantly associated with a higher risk of PCOS (OR = 6.22; 95% CI = 2.25, 17.16; $p$ = 0.0004). As per the WME method, a unit (1cm) increase in WC significantly increased PCOS risk (OR = 11.98; 95% CI = 3.30, 43.56; $p$ = 0.0002). According to the WMO method, a unit increase in WC conveyed an OR of 20.27 (95% CI = 2.32, 177.49; $p$ = 0.0152).

As revealed by the IVW method, a unit (1cm) increase in HC was significantly associated with a higher risk of PCOS (OR = 6.22; 95% CI = 2.25, 17.16; $p$ = 0.0004). According to the WME method, a unit increase in HC was associated with a significant increase in PCOS (OR = 11.98; 95% CI = 3.35, 42.81; $p$ = 0.0001). As per the WMO method, a unit increase in HC conveyed an OR of 20.27 (95% CI = 2.17, 189.65; $p$ = 0.0179).

However, MRE regression estimates did not achieve statistical significance with BMI, WC, and HC as exposures. Also, height and WHR were not significantly causally associated with PCOS in any 2SMR analyses.

## Method comparison plots

As shown in **Figs 1–4**, scatter plots and trend lines pertaining to the four significant exposures (weight, BMI, WC, HC) demonstrate positive causal associations with PCOS, according to

**Table 2. Results from two-sample Mendelian randomization analyses on the causality of anthropometric markers associated with polycystic ovarian syndrome.**

| Anthropometric marker | Method | No: of SNPs | β (SE) | 95% CI of β | p-value | OR | 95% CI of OR |
|---|---|---|---|---|---|---|---|
| Weight | **MR Egger** | **11** | **4.34 (1.63)** | **1.14, 7.54** | **0.0260** | **77.05** | **3.14, 1890.64** |
| | Weighted median | 11 | 0.55 (0.60) | -0.63, 1.73 | 0.3641 | 1.73 | 0.53, 5.64 |
| | **IVW** | **11** | **0.99 (0.48)** | **0.04, 1.94** | **0.0413** | **2.69** | **1.04, 6.95** |
| | Weighted mode | 11 | 0.33 (0.81) | -1.26, 1.92 | 0.6942 | 1.39 | 0.28, 6.79 |
| Height | MR Egger | 51 | -0.50 (0.81) | -2.09, 1.09 | 0.5426 | 0.61 | 0.12, 2.99 |
| | Weighted median | 51 | -0.23 (0.27) | -0.77, 0.30 | 0.3857 | 0.79 | 0.46, 1.34 |
| | IVW | 51 | -0.24 (0.20) | -0.63, 0.15 | 0.2216 | 0.78 | 0.53, 1.16 |
| | Weighted mode | 51 | -0.11 (0.52) | -1.14, 0.91 | 0.8280 | 0.89 | 0.32, 2.49 |
| **BMI** | MR Egger | 37 | 1.45 (0.79) | -0.09, 3.00 | 0.0733 | 4.28 | 0.91, 20.05 |
| | **Weighted median** | **37** | **1.06 (0.44)** | **0.20, 1.93** | **0.0162** | **2.90** | **1.22, 6.90** |
| | **IVW** | **37** | **1.40 (0.29)** | **0.83, 1.98** | **1.68E-06** | **4.06** | **2.29, 7.22** |
| | Weighted mode | 37 | 0.58 (0.75) | -0.90, 2.06 | 0.4448 | 1.79 | 0.41, 7.87 |
| WC | MR Egger | 17 | 3.35 (1.76) | -0.10, 6.79 | 0.0761 | 28.38 | 0.91, 886.82 |
| | **Weighted median** | **17** | **2.48 (0.66)** | **1.19, 3.77** | **0.0002** | **11.98** | **3.30, 43.56** |
| | **IVW** | **17** | **1.83 (0.52)** | **0.81, 2.84** | **0.0004** | **6.22** | **2.25, 17.16** |
| | **Weighted mode** | **17** | **3.01 (1.11)** | **0.84, 5.18** | **0.0152** | **20.27** | **2.32, 177.49** |
| HC | MR Egger | 17 | 3.35 (1.76) | -0.10, 6.79 | 0.0761 | 28.38 | 0.91, 886.82 |
| | **Weighted median** | **17** | **2.48 (0.65)** | **1.21, 3.76** | **0.0001** | **11.98** | **3.35, 42.81** |
| | **IVW** | **17** | **1.83 (0.52)** | **0.81, 2.84** | **0.0004** | **6.22** | **2.25, 17.16** |
| | **Weighted mode** | **17** | **3.01 (1.14)** | **0.77, 5.25** | **0.0179** | **20.27** | **2.17, 189.65** |
| WHR | MR Egger | 23 | 2.37 (1.92) | -1.39, 6.14 | 0.2304 | 10.73 | 0.25, 463.77 |
| | Weighted median | 23 | 0.79 (0.50) | -0.19, 1.77 | 0.1132 | 2.21 | 0.83, 5.90 |
| | IVW | 23 | 0.46 (0.36) | -0.25, 1.17 | 0.2001 | 1.59 | 0.78, 3.24 |
| | Weighted mode | 23 | 1.03 (0.69) | -0.33, 2.39 | 0.1512 | 2.80 | 0.72, 10.92 |

BMI = body mass index; HC = hip circumference; IVW = inverse variance weighted method; WC = waist circumference; WHR = waist to hip ratio.

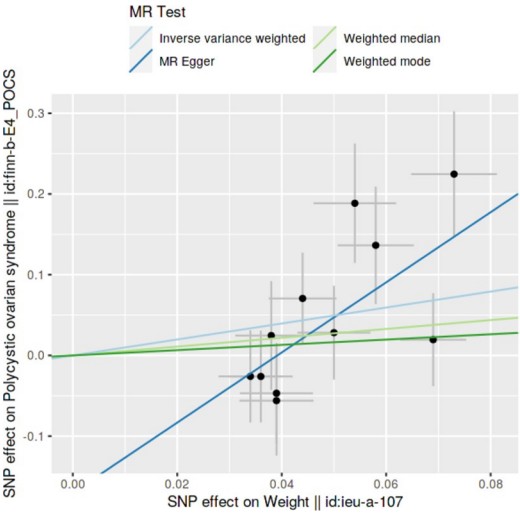

**Fig 1. Scatter plot illustrating the distribution of individual ratio estimates of weight with polycystic ovarian syndrome as the outcome.** Trend lines from the four different two-sample Mendelian randomization methods employed indicating the positive causal associations, are also included in each scatter plot.

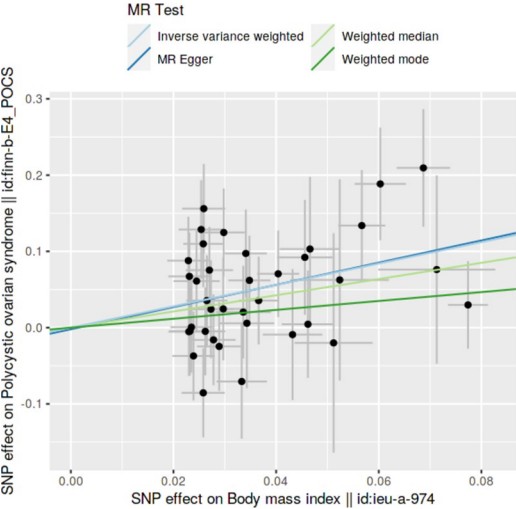

**Fig 2. Scatter plot illustrating the distribution of individual ratio estimates of body mass index with polycystic ovarian syndrome as the outcome.** Trend lines from the four different two-sample Mendelian randomization methods employed indicating the positive causal associations, are also included in each scatter plot.

different MR methods. Scatter plots and regression lines for the two non-significant exposures (height, WHR) are presented in **S1** and **S2** Figs.

## Single SNP analyses

Results from single SNP analyses are provided in **S3 Table**. Forest plots depicting single SNP analyses and pooled causal estimates as per IVW and MRE methods for significant exposures are presented in **Figs 5–8** while corresponding forest plots for non-significant exposures are given in **S3** and **S4** Figs. According to multiple testing corrected $p$-value thresholds, a single SNP ($rs2867131$; $p = 0.00394$; $p < 4.54 \times 10^{-3}$, 0.05/11) was individually associated with PCOS

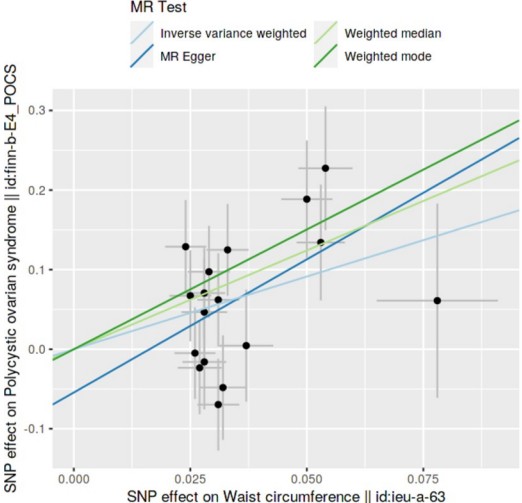

**Fig 3. Scatter plot illustrating the distribution of individual ratio estimates of waist circumference with polycystic ovarian syndrome as the outcome.** Trend lines from the four different two-sample Mendelian randomization methods employed indicating the positive causal associations, are also included in each scatter plot.

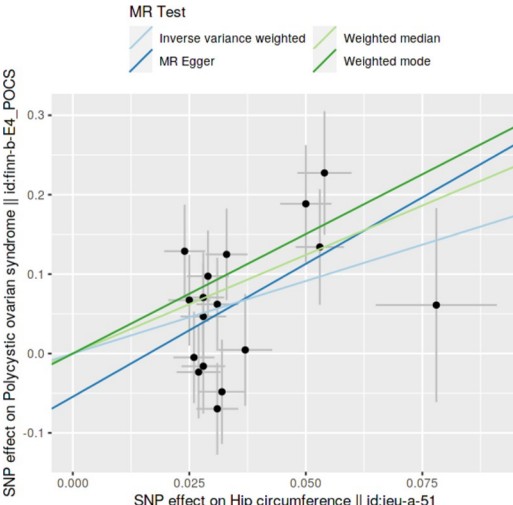

**Fig 4. Scatter plot illustrating the distribution of individual ratio estimates of hip circumference with polycystic ovarian syndrome as the outcome.** Trend lines from the four different two-sample Mendelian randomization methods employed indicating the positive causal associations, are also included in each scatter plot.

in the weight data. None of the SNPs were individually associated with PCOS in all other exposure data: height ($p > 9.80 \times 10^{-4}$, 0.05/51); BMI ($p > 1.35 \times 10^{-3}$; 0.05/37); WC and HC ($p > 2.94 \times 10^{-3}$; 0.05/17); WHR ($p > 2.17 \times 10^{-3}$; 0.05/23).

## Leave-one-out analyses

Results from leave-one-out analyses are presented in **S4 Table**. Leave-one-out analysis plots and pooled causal estimates as per IVW-MR method (for comparison) for significant exposures are presented in **Figs 9–12** while corresponding leave-one-out analysis plots for non-significant exposures are provided in **S5 and S6 Figs**. According to Bonferroni multiple testing corrected $p$-value thresholds, on exclusion of alternating SNPs, statistical significance was still

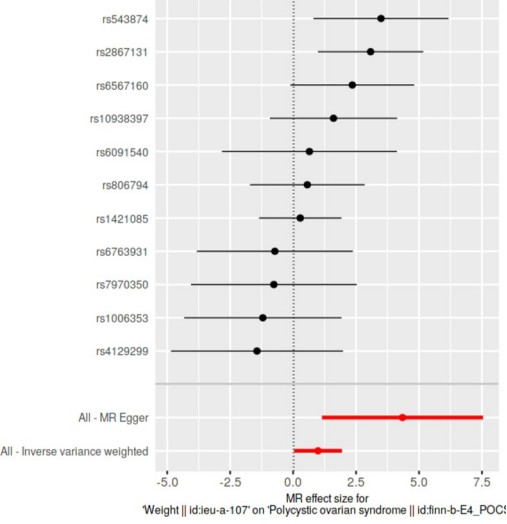

**Fig 5. Forest plot of weight, against polycystic ovarian syndrome as the outcome.** Effects of individual SNPs and pooled estimates from MR-Egger- and inverse variance weighted methods are visualized.

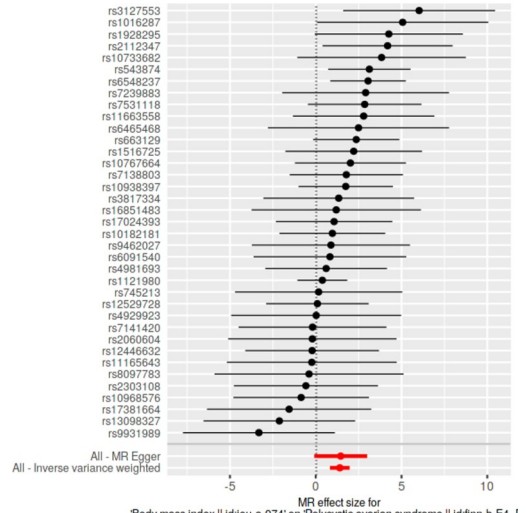

**Fig 6. Forest plot of body mass index, against polycystic ovarian syndrome as the outcome.** Effects of individual SNPs and pooled estimates from MR-Egger- and inverse variance weighted methods are visualized.

retained in three exposure datasets, indicating the robustness of their causal associations, despite the exclusion of any single SNP: BMI (p < 1.35 × 10–3; 0.05/37); WC and HC (p < 2.94 × 10–3; 0.05/17). Leave-one-out analyses of other anthropometric traits did not achieve statistical significance: weight ($p > 4.54 \times 10^{-3}$, 0.05/11); height ($p > 9.80 \times 10^{-4}$, 0.05/51); WHR ($p > 2.17 \times 10^{-3}$; 0.05/23).

## Heterogeneity analyses

Results from heterogeneity analyses (Cochran's Q- and *p*-values) are summarized in **Table 3**. We did not observe statistically significant heterogeneity in any 2SMR analyses ($p > 0.05$). Funnel plots for the significant exposures are illustrated in **Figs 13–16** while those for non-

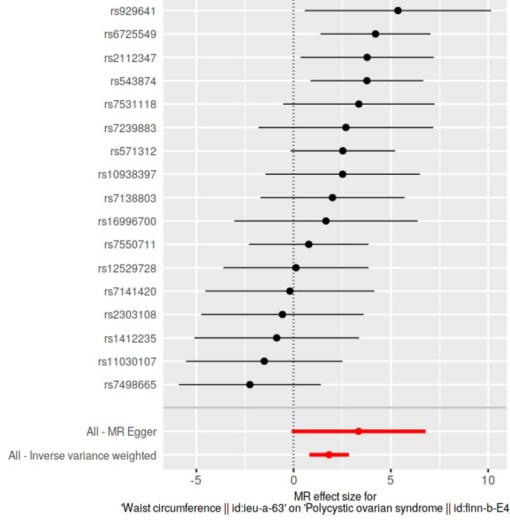

**Fig 7. Forest plot of waist circumference, against polycystic ovarian syndrome as the outcome.** Effects of individual SNPs and pooled estimates from MR-Egger- and inverse variance weighted methods are visualized.

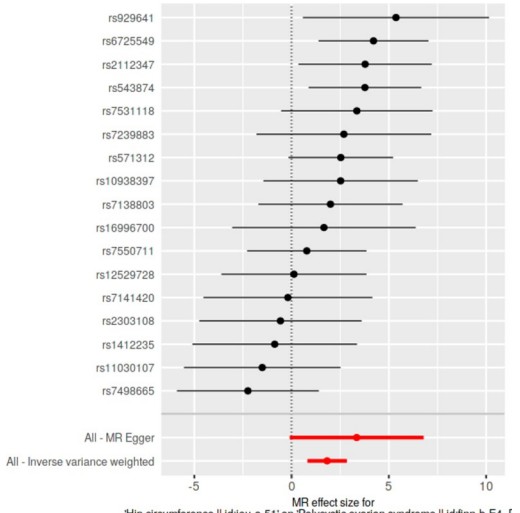

**Fig 8. Forest plot of hip circumference, against polycystic ovarian syndrome as the outcome.** Effects of individual SNPs and pooled estimates from MR-Egger- and inverse variance weighted methods are visualized.

significant exposures are provided in **S7** **and** **S8** **Figs**. In conformity with non-significant findings from heterogeneity analyses, asymmetric distributions indicating directional horizontal pleiotropy or large spreads suggesting considerable heterogeneity could not be discerned in funnel plots.

## Horizontal pleiotropy and outliers

We present the results of horizontal pleiotropy analyses via MRE regression intercepts and directionality $p$-values in **Table 4**. These analyses revealed that there was no significant horizontal pleiotropy ($p > 0.05$). As shown in **S5 Table**, congruent findings were obtained from

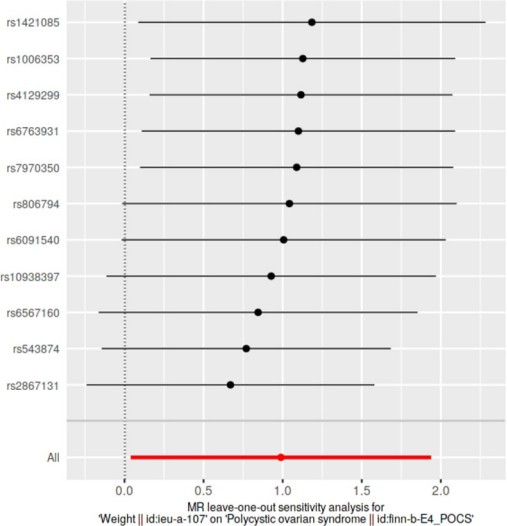

**Fig 9. Leave-one-out sensitivity analysis plot of weight, against polycystic ovarian syndrome as the outcome.** A given dark point indicates the effect measure from inverse variance weighted Mendelian randomization analysis excluding that specific SNP. The red lines indicate pooled analyses encompassing all SNPs via IVW-MR method (drawn for comparison).

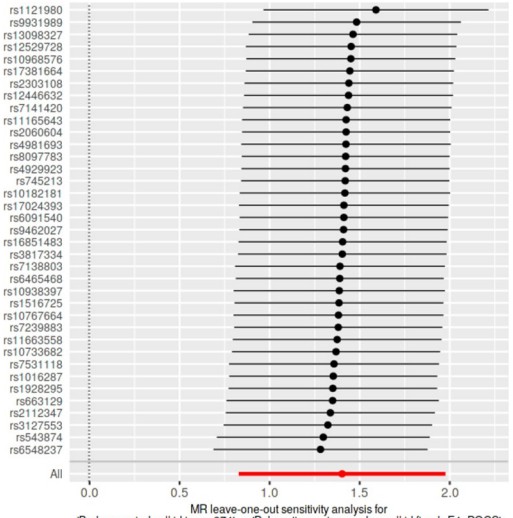

**Fig 10. Leave-one-out sensitivity analysis plot of body mass index, against polycystic ovarian syndrome as the outcome.** A given dark point indicates the effect measure from inverse variance weighted Mendelian randomization analysis excluding that specific SNP. The red lines indicate pooled analyses encompassing all SNPs via IVW-MR method (drawn for comparison).

MRPRESSO analyses which confirmed the absence of outliers and no significant horizontal pleiotropy (global test $p$-values > 0.05). Radial plots for significant exposures are provided in **Figs 17–20** while those for non-significant exposures are given in **S9 and S10 Figs**. All radial plots illustrate the absence of outliers.

## Genetic architectures underlying causality

**Table 5** presents SNPs and nearest genes/TSSs underlying the four significant causal associations: weight, BMI, WC and HC versus PCOS. Notably, both WC and HC had the same 17

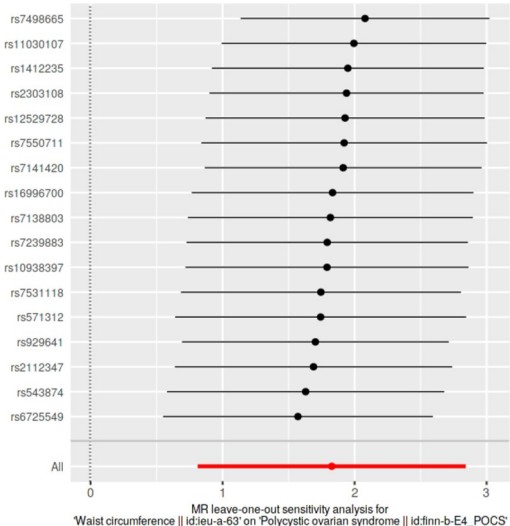

**Fig 11. Leave-one-out sensitivity analysis plot of waist circumference, against polycystic ovarian syndrome as the outcome.** A given dark point indicates the effect measure from inverse variance weighted Mendelian randomization analysis excluding that specific SNP. The red lines indicate pooled analyses encompassing all SNPs via IVW-MR method (drawn for comparison).

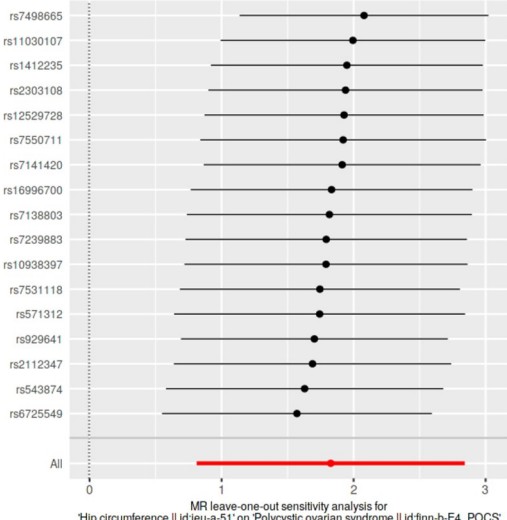

**Fig 12. Leave-one-out sensitivity analysis plot of hip circumference, against polycystic ovarian syndrome as the outcome.** A given dark point indicates the effect measure from inverse variance weighted Mendelian randomization analysis excluding that specific SNP. The red lines indicate pooled analyses encompassing all SNPs via IVW-MR method (drawn for comparison).

SNPs underlying their significant causal associations with PCOS. There were 11 SNPs underlying the significant causal association between weight and PCOS and 37 SNPs underlying the association between BMI and PCOS. Two common SNPs (*rs10938397, rs543874*) were observed across these four associations. The number of genes underlying the associations of weight, BMI, and WC/HC versus PCOS were 11, 40, and 21, respectively. Of these, 5 genes (*GNPDA2, MC4R, SEC16B, TMEM18, ZFP64*) were shared across the four significant associations. Of the 11 neighboring genes underlying the weight-PCOS association, only 4 (*H2BC7, HMGA2, MTIF3, ZBTB38*) were exclusive/non-shared. Of the 40 neighboring genes underlying the BMI-PCOS association, only 19 (*ADCY3, AMPD2, BEND5, ETV5, FOXG1, FTO, GPRC5B, HNF4G, LMX1B, MTCH2, NPC1, PDK4, PTBP2, RASA2, SKOR1, STK33, TLR4,*

**Table 3. Heterogeneity statistics of two-sample Mendelian randomization analyses.**

| Anthropometric marker | Method | Q value | Degrees of freedom | *p*-value |
|---|---|---|---|---|
| Weight | MR Egger | 10.35 | 9 | 0.3226 |
| | IVW | 15.55 | 10 | 0.1132 |
| Height | MR Egger | 56.11 | 49 | 0.2257 |
| | IVW | 56.23 | 50 | 0.2529 |
| BMI | MR Egger | 34.11 | 35 | 0.5107 |
| | IVW | 34.12 | 36 | 0.5583 |
| Waist circumference | MR Egger | 20.16 | 15 | 0.1659 |
| | IVW | 21.26 | 16 | 0.1687 |
| Hip circumference | MR Egger | 20.16 | 15 | 0.1659 |
| | IVW | 21.26 | 16 | 0.1687 |
| Waist to hip ratio | MR Egger | 16.11 | 21 | 0.7634 |
| | IVW | 17.14 | 22 | 0.7559 |

BMI = body mass index; IVW = inverse variance weighted method.

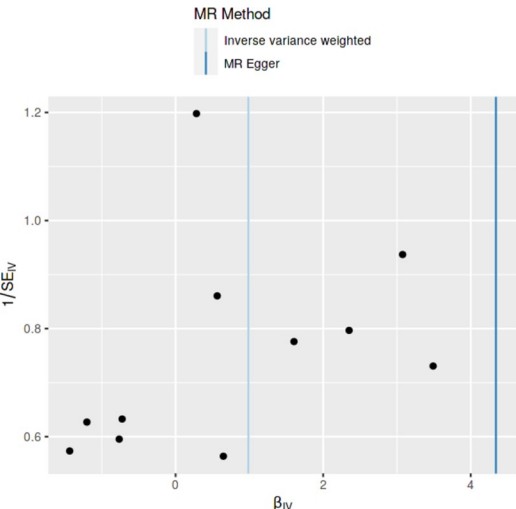

**Fig 13. Funnel plot of weight as the exposure, against polycystic ovarian syndrome as the outcome.**

*UHRF1BP1, ZZZ3*) were exclusive/non-shared. Of the 21 neighboring genes underlying the associations between WC/HC and PCOS, only 2 (*GPR61, VRK2*) were exclusive.

## Multivariable MR findings

Results of multivariable MR analyses are summarized in **Table 6**. No significant pleiotropic effects were observed in all multivariable MR models, as indicated by non-significant heterogeneity statistic ($p > 0.05$). Of the seven models assessed, only two models containing exclusively the anthropometric markers of central obesity produced significant direct causal effects: multivariable MR model 2 containing HC and WC as exposures and multivariable MR model 7 containing HC, WC, and WHR as exposures. In both cases, HC and WC emerged significant: in multivariable MR model 2, $\beta = 1.827$ and $p = 0.00042$ for both WC and HC; in multivariable MR model 7, $\beta = 2.001$ and $p = 0.001$, for both WC and HC. Other five multivariable models

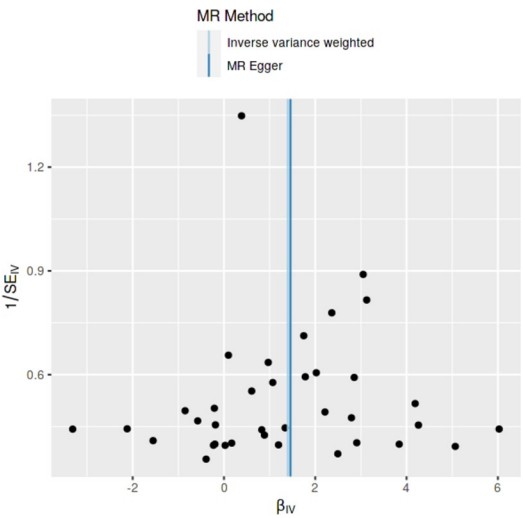

**Fig 14. Funnel plot of body mass index as the exposure, against polycystic ovarian syndrome as the outcome.**

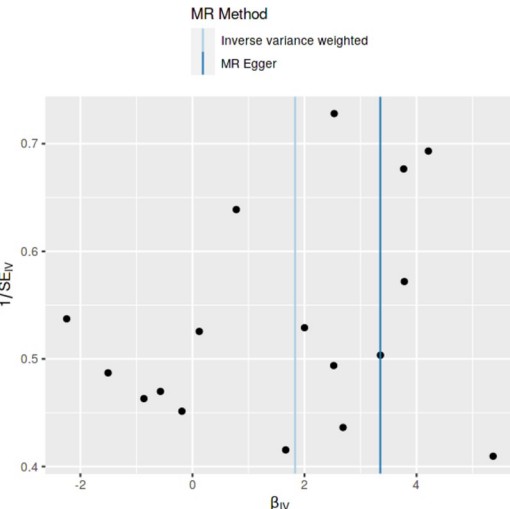

**Fig 15. Funnel plot of waist circumference as the exposure, against polycystic ovarian syndrome as the outcome.**

which contained surrogate anthropometric markers of both central- and general- obesity as exposures demonstrated no statistical significance, indicating lack of direct causal effects on PCOS in these models. Clinical implications of these findings alluding to the complexity of causal links between central- and general- obesity versus PCOS have been elaborated in Discussion.

## Bidirectional MR findings

Bidirectional MR analyses taking PCOS as the exposure and each anthropometric trait as the outcome retrieved only a single functional variant/SNP (*rs1531788*) against the specified criteria: genome-wide significance threshold $p < 5e - 08$; clumping LD-$R^2 < 0.001$; clumping distance $> 10000$ kb; proxy SNPs via LD-tagging with minimum LD-$R^2 > 0.8$; MAF threshold for aligning palindromic SNPs = 0.3. Therefore, we could not reliably assess causal associations

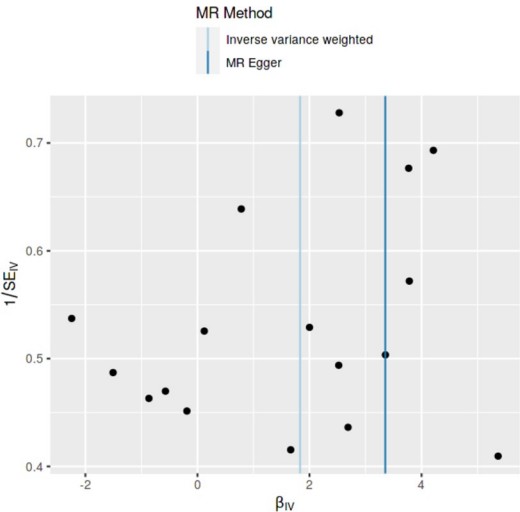

**Fig 16. Funnel plot of hip circumference as the exposure, against polycystic ovarian syndrome as the outcome.**

**Table 4. Horizontal pleiotropy statistics of two-sample Mendelian randomization analyses.**

| Anthropometric marker | Egger regression intercept | Standard error | Directionality *p*-value |
|---|---|---|---|
| Weight | -0.17 | 0.08 | 0.0625 |
| Height | 0.012 | 0.039 | 0.748 |
| Body mass index (BMI) | -0.002 | 0.029 | 0.944 |
| Waist circumference (WC) | -0.054 | 0.06 | 0.379 |
| Hip circumference (HC) | -0.054 | 0.06 | 0.379 |
| Waist to hip ratio (WHR) | -0.07 | 0.069 | 0.323 |

in the opposite direction using our data sources. In Discussion section, we contextualized findings from previous bidirectional MR studies which had evaluated the reverse causality between obesity and PCOS.

## Discussion

To our knowledge, this is the first MR study to comprehensively and systematically analyze total and direct causal associations between surrogate anthropometric markers of both general- and central- obesity and PCOS, using large-sample, female-only cohorts of European ancestry. Since none of the previous studies assessing causality between obesity and PCOS conducted multivariable MR analyses to shed light on direct causal effects [25–28], our study offers novel and more profound insights, compared to prior research. We further report shared and non-shared genetic architectures with respect to potentially causal anthropometric traits in ensuing PCOS.

### Findings from univariable 2SMR models: Total causal effects

We found that both central- and general- obesity potentially yield overall, total causal effects on PCOS, as signified by the four significant anthropometric markers–weight, BMI, WC, and HC, as per univariable 2SMR models. With respect to general obesity, our findings are congruent with previous 2SMR analyses which reported positive total causal effects of BMI on PCOS [25–28]. With regard to central obesity also, we observed consistently positive total causal

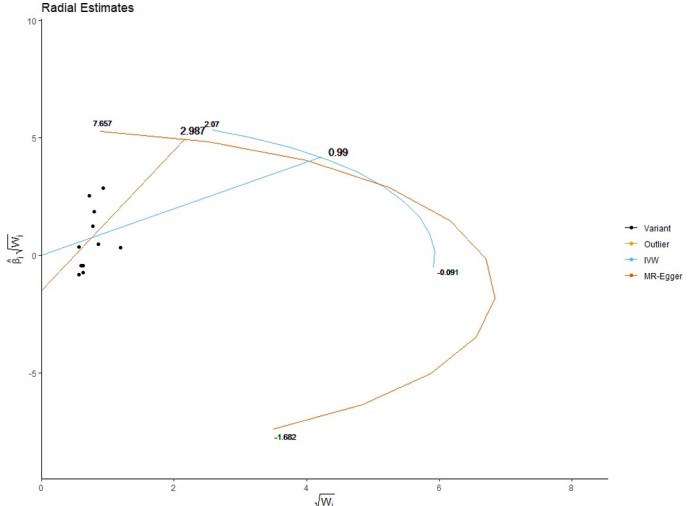

**Fig 17. Radial plot of weight as the exposure.** No significant outliers were detected.

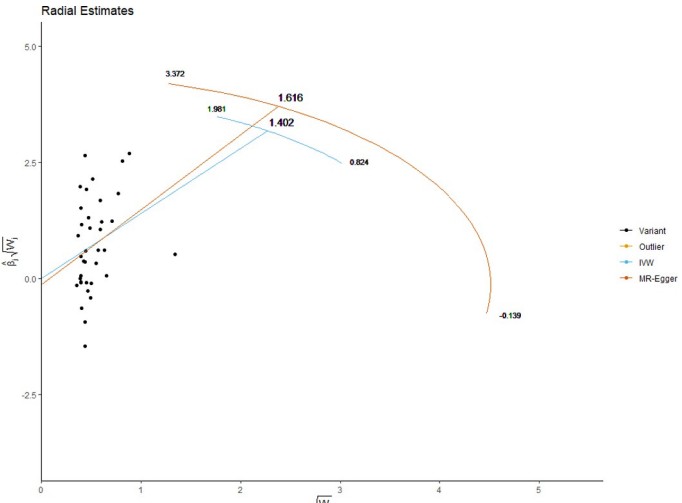

**Fig 18. Radial plot of body mass index as the exposure.** No significant outliers were detected.

effects of WC and HC on PCOS, as indicated by significant causal estimates across a number of univariable 2SMR models (OR range: 6.22–20.27). With regard to all four markers–weight, BMI, WC, and HC, we observed significance from more than a single 2SMR model, which was reassuring. Specifically, the standard IVW method as well as one or more robust methods (MRE, WME, WMO) demonstrated statistical significance, providing consistent evidence on significant total causal effects of both central- and general obesity in ensuing PCOS (**Table 2**).

We ensured minimal overlap between exposure-outcome samples–a key requirement for reducing bias in 2SMR estimates [14]. In fact, non-substantial sample overlaps, especially involving controls, are even permissible, and would not evoke bias in 2SMR studies [53]. No confounding by ancestry entailed as all participants were of European ancestry. However, this may have led to a lack of generalizability of our findings and transethnic validations via MR in independent cohorts would be required. Further, there was no weak instrument bias as all IVs had F > 10. Robustness of our findings was underscored by a range of sensitivity analyses,

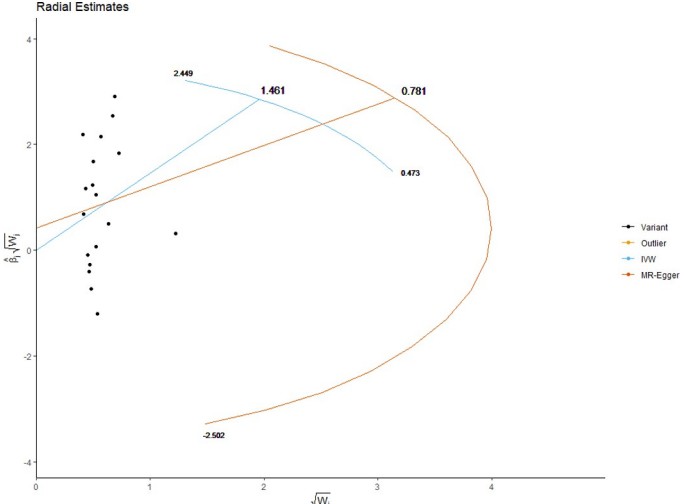

**Fig 19. Radial plot of waist circumference as the exposure.** No significant outliers were detected.

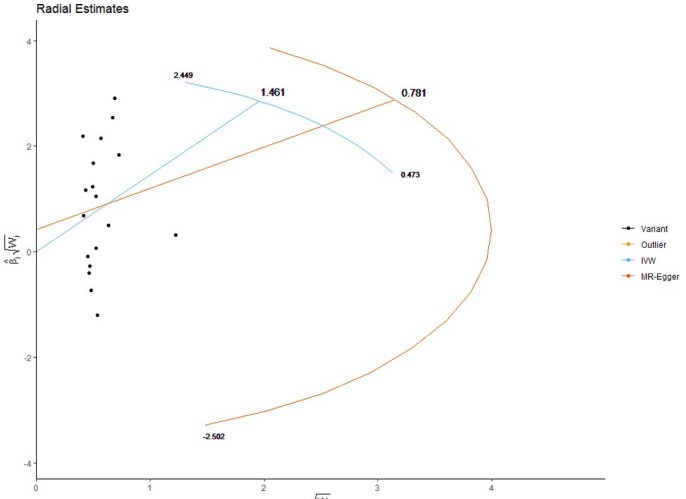

**Fig 20. Radial plot of hip circumference as the exposure.** No significant outliers were detected.

which uncovered no substantial heterogeneity, horizontal pleiotropy, and outliers. While the use of summarized public data has advantages such as transparency, reproducibility, and the greater ability to detect causal associations due to large samples, it has a few shortcomings as well. For example, we could not conduct subgroup analyses, rule out collider bias due to over-adjusted summarized association estimates, or assess potentially non-linear associations of obesity with PCOS, for which individual-level data are required [38]. The relatively small number of PCOS cases in the outcome GWAS (N = 642) might be a limiting factor, given the community prevalence of PCOS > 8% [2], as opposed to the substantially lower prevalence of clinically-diagnosed cases (~0.54%) in the Finnish cohort used in the present study. However, these are women with overt PCOS presenting to the clinicians with common, cardinal clinical features including obesity and predominant PCOS phenotypes, as confirmed by ICD codes. As we uncovered potential causalities despite the small number of cases, stronger findings with larger causal estimates might possibly be obtained from independent cohorts containing a larger number of women with PCOS, which need to be investigated in future studies. For future studies, we suggest meta-analyzing several cohorts from sources such as the UK Bio-bank and FinnGen as a judicious avenue for formidably increasing the sample size. Although we minimized confounding with female-only cohorts of European ancestry and summary estimates adjusted for multiple variables, there may still have been residual confounding influencing our causal estimates toward or away from the null.

## Genetics underlying causality

We observed a considerable degree of overlap across significant causal findings, perhaps indicating similar genetically-driven causal mechanisms. However, there were SNPs and genes that were exclusive to each causal association as well, suggesting unique, exposure-specific pathways may also participate in PCOS pathogenesis. This suggests that differences may exist in the genetic basis of PCOS among women with central obesity alone (normal weight-central obesity), as compared to women with both general- and central obesity. Of the identified genes in our study, *FTO*, *GNPDA2*, *MC4R*, *MTCH2*, *NEGR1*, *SH2B1*, *TMEM18* are unequivocally regarded as obesity genes associated with PCOS, as supported by contemporary evidence [54]. A gene set rather similar to those found in the present study (*ADCY3*, *ATP2A1*, *BCDIN3D*,

**Table 5. Genetic variants (SNPs) and nearest genes/transcriptional start sites (TSSs) underlying the significant causal associations.**

| Anthropometric marker | SNP | Effect allele (exposure) | Other allele (exposure) | Effect allele (Outcome) | Other allele (Outcome) | Nearest gene(s)/TSSs |
|---|---|---|---|---|---|---|
| Weight | *rs1006353* | G | A | G | A | *MTIF3* |
| | ***rs10938397*** | G | A | G | A | ***GNPDA2*** |
| | *rs1421085* | C | T | C | T | ***RPGRIP1L*** |
| | *rs2867131* | C | T | C | T | ***TMEM18*** |
| | *rs4129299* | G | A | G | A | ***CADM2*** |
| | ***rs543874*** | G | A | G | A | ***SEC16B*** |
| | ***rs6091540*** | T | C | T | C | ***ZFP64*** |
| | *rs6567160* | C | T | C | T | ***MC4R*** |
| | *rs6763931* | A | G | A | G | ZBTB38 |
| | *rs7970350* | T | C | T | C | HMGA2 |
| | *rs806794* | G | A | G | A | H2BC7 |
| BMI | *rs1016287* | C | T | C | T | ***FANCL*** |
| | *rs10182181* | G | A | G | A | ADCY3 |
| | *rs10733682* | G | A | G | A | LMX1B |
| | *rs10767664* | A | T | A | T | ***BDNF*** |
| | ***rs10938397*** | G | A | G | A | ***GNPDA2*** |
| | *rs10968576* | G | A | G | A | ***LINGO2*** |
| | *rs11165643* | T | C | T | C | PTBP2 |
| | *rs1121980* | A | G | A | G | FTO; *RPGRIP1L* |
| | *rs11663558* | A | G | A | G | NPC1 |
| | *rs12446632* | A | G | A | G | GPRC5B |
| | ***rs12529728*** | G | A | G | A | ***TFAP2B*** |
| | *rs13098327* | A | G | A | G | ***CADM2*** |
| | *rs1516725* | C | T | C | T | ETV5 |
| | *rs16851483* | T | G | T | G | RASA2 |
| | *rs17024393* | C | T | C | T | AMPD2 |
| | *rs17381664* | C | T | C | T | ZZZ3 |
| | *rs1928295* | C | T | C | T | TLR4 |
| | *rs2060604* | C | T | C | T | HNF4G |
| | ***rs2112347*** | G | T | G | T | ***POC5*** |
| | ***rs2303108*** | C | T | C | T | ***SAE1; ZC3H4*** |
| | *rs3127553* | A | G | A | G | BEND5 |
| | *rs3817334* | T | C | T | C | MTCH2 |
| | *rs4929923* | C | T | C | T | STK33 |
| | *rs4981693* | A | G | A | G | FOXG1 |
| | ***rs543874*** | G | A | G | A | ***SEC16B*** |
| | ***rs6091540*** | T | C | T | C | ***ZFP64*** |
| | *rs6465468* | T | G | T | G | PDK4 |
| | *rs6548237* | C | A | C | A | ***TMEM18*** |
| | *rs663129* | A | G | A | G | ***MC4R*** |
| | ***rs7138803*** | A | G | A | G | ***BCDIN3D; NCKAP5L*** |
| | ***rs7141420*** | T | C | T | C | ***DIO2*** |
| | ***rs7239883*** | A | G | A | G | ***RIT2*** |
| | *rs745213* | G | T | G | T | SKOR1 |
| | ***rs7531118*** | C | T | C | T | ***NEGR1*** |
| | *rs8097783* | A | G | A | G | ***MC4R*** |

*(Continued)*

Table 5. (Continued)

| Anthropometric marker | SNP | Effect allele (exposure) | Other allele (exposure) | Effect allele (Outcome) | Other allele (Outcome) | Nearest gene(s)/TSSs |
|---|---|---|---|---|---|---|
| | rs9462027 | A | G | A | G | UHRF1BP1 |
| | rs9931989 | C | G | C | G | ATP2A1; SH2B1 |
| WC/HC* | rs10938397 | G | A | G | A | GNPDA2 |
| | rs11030107 | G | A | G | A | BDNF |
| | rs12529728 | G | A | G | A | TFAP2B |
| | rs1412235 | C | G | C | G | LINGO2 |
| | rs16996700 | C | T | C | T | ZFP64 |
| | rs2112347 | G | T | G | T | POC5 |
| | rs2303108 | C | T | C | T | SAE1; ZC3H4 |
| | rs543874 | G | A | G | A | SEC16B |
| | rs571312 | A | C | A | C | MC4R |
| | rs6725549 | A | C | A | C | TMEM18 |
| | rs7138803 | A | G | A | G | BCDIN3D; NCKAP5L |
| | rs7141420 | T | C | T | C | DIO2 |
| | rs7239883 | A | G | A | G | RIT2 |
| | rs7498665 | G | A | G | A | ATP2A1; SH2B1 |
| | rs7531118 | C | T | C | T | NEGR1 |
| | rs7550711 | T | C | T | C | GPR61 |
| | rs929641 | G | A | G | A | FANCL; VRK2 |

*WC and HC demonstrated the same genetic architecture (17 SNPs/21 genes) underlying their significant causal associations with polycystic ovarian syndrome. Shared/overlapping SNPs and nearest genes/TSSs across anthropometric markers are highlighted in bold.

BMI = body mass index; HC = hip circumference; WC = waist circumference.

BDNF, CADM2, ETV5, FTO, GNPDA2, GPRC5B, HNF4G, LINGO2, LMX1B, MC4R, MTCH2, NEGR1, POC5, PTBP2, RASA2, SEC16B, TFAP2B, TLR4, TMEM18, ZC3H4, ZFP64) was underlying the causal association between BMI and PCOS in a previous MR study as well [25].

Mechanisms underlying the causal effects of central-and general obesity on PCOS are not fully understood and need to be investigated. Towards this end, the SNPs and neighboring genes underlying causal associations uncovered by significant 2SMR models in the present study may be insightful. In S6 Table, we have presented congruent evidence from other studies supporting for putative mechanistic roles of these shared and non-shared genes underlying the causal associations between anthropometric markers of central- and/or general obesity and PCOS. However, it was beyond the scope of the current study to explore pathophysiological functions of these genes in the development of PCOS, and we recommend their putative pathobiological relevance for PCOS is explored in future experiments.

There is a formidable body of evidence supporting FTO gene's association with both obesity and PCOS. While FTO is, in fact, the first locus identified as unequivocally associated with adiposity [55], a meta-analysis revealed a direct association between FTO variant and PCOS risk, independent of BMI [56]. Both FTO and MC4R gene variants are associated with obesity in PCOS [57] while observational evidence indicates a direct role of the interaction between FTO and MC4R polymorphisms in the development of PCOS [58]. TLR4 and toll-like receptor genes in general contribute to the development of chronic low-grade inflammation as well as insulin resistance and hyperandrogenism observed in PCOS [59] whereas a MR study revealed essential causal roles of systemic inflammatory regulators, especially cytokines, in the

**Table 6. Findings from multivariable Mendelian randomization analyses with multiple anthropometric markers as exposures and polycystic ovarian syndrome as the outcome.**

| Exposure | nSNP | β | SE | 95% CI of β | p-value | Residual SE | Heterogeneity |
|---|---|---|---|---|---|---|---|
| **Multivariable MR model 1: HC + WC + BMI** | | | | | | | |
| HC | 18 | -0.453 | 2.490 | -5.333, 4.426 | 0.856 | 1.014 | Cochran's Q = 35.9749 |
| WC | 18 | -0.453 | 2.490 | -5.333, 4.426 | 0.856 | | p-value = 0.4227 |
| BMI | 37 | 1.842 | 2.161 | -2.394, 6.078 | 0.394 | | |
| **Multivariate MR model 2: HC + WC** | | | | | | | |
| **HC** | **17** | **1.827** | **0.518** | **0.812, 2.842** | **0.00042** | **1.153** | Cochran's Q = 19.9327 |
| **WC** | **17** | **1.827** | **0.518** | **0.812, 2.842** | **0.00042** | | p-value = 0.1745 |
| **Multivariable MR model 3: WC + BMI** | | | | | | | |
| WC | 18 | -0.453 | 2.490 | -5.333, 4.426 | 0.856 | 1.014 | Cochran's Q = 37.0027 |
| BMI | 37 | 1.842 | 2.161 | -2.394, 6.078 | 0.394 | | p-value = 0.4225 |
| **Multivariable MR model 4: HC + BMI** | | | | | | | |
| HC | 18 | -0.453 | 2.490 | -5.333, 4.426 | 0.856 | 1.014 | Cochran's Q = 37.0027 |
| BMI | 37 | 1.842 | 2.161 | -2.394, 6.078 | 0.394 | | p-value = 0.4225 |
| **Multivariable MR model 5: HC + Weight** | | | | | | | |
| HC | 17 | -0.532 | 1.253 | -2.989, 1.924 | 0.671 | 1.141 | Cochran's Q = 26.0236 |
| Weight | 9 | 2.187 | 1.509 | -0.771, 5.145 | 0.147 | | p-value = 0.1650 |
| **Multivariate MR model 6: WC + Weight** | | | | | | | |
| WC | 17 | -0.532 | 1.253 | -2.989, 1.924 | 0.671 | 1.141 | Cochran's Q = 26.0236 |
| Weight | 9 | 2.187 | 1.509 | -0.771, 5.145 | 0.147 | | p-value = 0.1650 |
| **Multivariate MR model 7: HC + WC + WHR** | | | | | | | |
| **HC** | **15** | **2.001** | **0.628** | **0.771, 3.232** | **0.001** | **1.030** | Cochran's Q = 35.0249 |
| **WC** | **15** | **2.001** | **0.628** | **0.771, 3.232** | **0.001** | | p-value = 0.3722 |
| WHR | 22 | -0.294 | 0.499 | -1.273, 0.684 | 0.556 | | |

BMI = body mass index; CI = confidence interval; HC = hip circumference; nSNP = number of SNPs; SE = standard error; WC = waist circumference; WHR = waist to hip ratio.

pathogenesis of PCOS [60]. However, mechanisms by which certain other genes found to be underlying the causal associations in the present study contribute to the development of PCOS, are not completely known.

Current evidence indicates that obesity contributes to the onset and progression of PCOS through multiple mechanisms including the escalation of insulin resistance and compensatory hyperinsulinemia which in turn leads to augmented adipogenesis and reduced lipolysis, sensitization of thecal cells to luteinizing hormone and intensification of functional ovarian hyperandrogenism by increasing ovarian androgen synthesis, and elevation of inflammatory adipokines which in turn leads to upregulation of insulin resistance and adipogenesis [19]. Indeed, obesity contributes to the pathogenesis of PCOS via multiple mechanisms that encompass the three cardinal clinical manifestations of PCOS, namely, hyperandrogenism, reproductive-, and metabolic dysfunction [61]. Moreover, our findings allude to the concept of secondary PCOS, which has only emerged in recent years. It has been proposed that adiposity and hyperinsulinemia (exogenous in type 1- and endogenous in type 2- diabetes) cause PCOS [62]. Recent research on polygenic risks scores has also demonstrated that both men and women manifest similar metabolic features of PCOS and that the presence of ovaries is neither essential nor required for PCOS while obesity features prominently [63]. Genetic research is providing greater insights into the mechanisms and nature of this complex polygenic disorder.

Given the differences in hormonal and metabolic profiles, treatment strategies and outcomes between obese- and non-obese PCOS phenotypes, it has been proposed to incorporate obesity as a clinical sign for classifying PCOS phenotypes [64]. However, lean PCOS phenotypes which are less prevalent than obese or overweight PCOS phenotypes, may have different underlying genetic architectures and pathogenic mechanisms and may need to be investigated separately.

## Findings of multivariable MR models and their implications

Results from multivariable MR modelling allude to the complex nature of causal associations between obesity and PCOS. As suggested by the two significant multivariable MR models which comprised of only the anthropometric markers of central obesity, it is possible that in the presence of normal-weight central obesity, direct causal effects are exerted on the onset of PCOS. However, in the presence of both central- and general- obesity, more complex causal mechanisms seem to be at play, as indicated by all corresponding, non-significant multivariable MR models. These findings strongly suggest the possible existence of intricate phenomena such as causal mediation when both central- and general obesity are present. For instance, a separate study which found total causal effects of both sex hormone-binding globulin and testosterone on coronary heart disease, but no direct causal effects, led to the discovery of causal mediations of both these exposures on coronary heart disease via subsequent causal network MR modelling [65]. Although we did not perform network MR, such downstream analyses will be useful to dissect complex causal mechanisms between obesity and PCOS. Other potential phenomena such as nonlinear causality and causal interactions should also be assessed in future studies with prudent study designs such as nonlinear MR [66] and factorial MR [67], respectively. Lastly, other caveats such as causality between traits and high polygenicity of traits should also be assessed in future studies, to fully disentangle complex causal mechanisms between obesity and PCOS.

## Bidirectional associations/reverse causality

For bidirectional 2SMR analyses, sufficient number of IVs could not be retrieved from existing data sources, but, with the gradual expansion of these open-source GWAS summary data, we envisage that it will be possible to conduct bidirectional analyses more robustly, in the future. Stringent eligibility criteria to select only highly uncorrelated and independent SNPs (LD-$R^2$ = 0.001) may also have contributed to the lack of functional variants produced for assessing reverse causality. However, concordant evidence from previous bidirectional analyses provides reassurance that there is no reverse causality i.e. PCOS would not cause obesity [25, 26].

## Clinical and public health value of our findings

Our findings indicating complex causal associations of obesity and adiposity with PCOS have clinical and public health implications. This analysis reinforced findings of previous studies indicating total causal effects of general obesity on PCOS [25–28]. In addition, this study offered novel results such as the total causal effects of central obesity on PCOS, potentially direct causal effects of normal weight-central obesity on PCOS, and that complex causal mechanisms in the presence of both general- and central obesity might lead to PCOS.

Of note, although our analyses indicated possibly direct causal effects of normal weight-central obesity on PCOS, people with central obesity but of normal weight are generally ignored in clinical guidelines [68]. Therefore, women with normal weight-central obesity will require an equally important focus as attributed to general obesity phenotype, in clinical guidelines for PCOS management. To this end, Mediterranean style dietary patterns and physical activity both are evidence-based interventions to reduce central obesity and ensure weight stabilization [69].

Causality of central obesity/adiposity demarcated by WC/HC in the present study corroborate findings from mechanistic studies focused on PCOS etiology. Truncal/visceral obesity promotes insulin resistance which in turn aggravates metabolic and reproductive functions in PCOS such as hyperandrogenism, dyslipidemia, and anovulation [70]. Evidence suggests the formation of a vicious cycle by central obesity which involves hyperandrogenemia and truncal adiposity in PCOS [71].

Non-significance of WHR might perhaps be owing to its lesser validity and reliability as an anthropometric marker of central obesity, compared to WC/HC, as concluded by previous studies [72, 73]. Lack of adequate statistical power in MR models to uncover causal associations from open-source data might be another contributory factor. This may be verified in future studies using larger cohorts.

On the whole, findings of this study indicating complex causality of central- and general obesity on PCOS, highlight the need to address the prevention of both forms of obesity at a population level through successful fiscal policies and statutory regulations [74]. Our findings are also congruent with recommendations from current international guidelines on PCOS, which underscore the prevention of excess weight gain in all women with PCOS, alongside lifestyle interventions focused on diet, weight loss, and physical activity for women with PCOS with overweight and obesity. As in the standard clinical management of obesity, pharmacotherapy and other treatments may also be appropriate [19].

## Conclusions

In this study, we revealed that both central- and general obesity yield total causal effects on PCOS, via 2SMR analyses using female-only, large-sample cohorts of European ancestry. Furthermore, it demonstrated potential direct causal effects of normal weight-central obesity on PCOS and complex causal mechanisms associated with PCOS in the presence of both central- and general obesity. Findings warrant further studies to explore genetic mechanisms underpinning causal associations of obesity with this common and complex condition. They also support international guidelines and underscore the importance of addressing both central- and general obesity phenotypes and adiposity for the prevention and management of PCOS.

## Supporting information

**S1 Fig. Scatter plot illustrating the distribution of individual ratio estimates of height as the exposure, with polycystic ovarian syndrome as the outcome.** Trend lines from the four different two-sample Mendelian randomization methods employed, are also included in each scatter plot.
(TIF)

**S2 Fig. Scatter plot illustrating the distribution of individual ratio estimates of waist-to-hip ratio as the exposure, with polycystic ovarian syndrome as the outcome.** Trend lines from the four different two-sample Mendelian randomization methods employed, are also included in each scatter plot.
(TIF)

**S3 Fig. Forest plot of height as the exposure, against polycystic ovarian syndrome as the outcome.** Effects of individual SNPs and pooled estimates from MR-Egger- and inverse variance weighted methods are visualized.
(TIF)

**S4 Fig. Forest plot of waist-to-hip ratio as the exposure, against polycystic ovarian syndrome as the outcome.** Effects of individual SNPs and pooled estimates from MR-Egger- and

inverse variance weighted methods are visualized.
(TIF)

**S5 Fig. Leave-one-out sensitivity analysis plot of height as the exposure, against polycystic ovarian syndrome as the outcome.** A given dark point indicates the effect measure from inverse variance weighted Mendelian randomization analysis excluding that specific SNP. The red lines indicate pooled analyses encompassing all SNPs.
(TIF)

**S6 Fig. Leave-one-out sensitivity analysis of the plot of waist-to-hip ratio as the exposure, against polycystic ovarian syndrome as the outcome.** A given dark point indicates the effect measure from inverse variance weighted Mendelian randomization analysis excluding that specific SNP. The red lines indicate pooled analyses encompassing all SNPs.
(TIF)

**S7 Fig. Funnel plot of height as the exposure against polycystic ovarian syndrome as the outcome.**
(TIF)

**S8 Fig. Funnel plot of waist-to-hip ratio as the exposure against polycystic ovarian syndrome as the outcome.**
(TIF)

**S9 Fig. Radial plot of height as the exposure.** No significant outliers were detected.
(TIF)

**S10 Fig. Radial plot as waist-to-hip ratio as the exposure.** No significant outliers were detected.
(TIF)

**S1 Table. STROBE-MR checklist.**
(DOCX)

**S2 Table. Harmonized data.**
(XLSX)

**S3 Table. Single SNP analyses.**
(XLSX)

**S4 Table. Leave-one-out analyses.**
(XLSX)

**S5 Table. MR PRESSO analyses.**
(DOCX)

**S6 Table. Congruent evidence from other studies supporting for putative mechanistic roles of the shared and non-shared genes underlying the causal associations between anthropometric markers of central—and/or general obesity and PCOS.**
(DOCX)

## Author Contributions

**Conceptualization:** Kushan De Silva, Ryan T. Demmer, Daniel Jönsson, Aya Mousa, Joanne Enticott.

**Data curation:** Kushan De Silva.

**Formal analysis:** Kushan De Silva.

**Funding acquisition:** Kushan De Silva.

**Investigation:** Kushan De Silva.

**Methodology:** Kushan De Silva, Ryan T. Demmer, Daniel Jönsson, Aya Mousa, Helena Teede, Andrew Forbes, Joanne Enticott.

**Project administration:** Kushan De Silva, Ryan T. Demmer, Joanne Enticott.

**Resources:** Kushan De Silva, Ryan T. Demmer, Joanne Enticott.

**Software:** Kushan De Silva.

**Supervision:** Ryan T. Demmer, Daniel Jönsson, Aya Mousa, Helena Teede, Andrew Forbes, Joanne Enticott.

**Validation:** Ryan T. Demmer, Daniel Jönsson, Aya Mousa, Helena Teede, Andrew Forbes, Joanne Enticott.

**Visualization:** Kushan De Silva.

**Writing – original draft:** Kushan De Silva.

**Writing – review & editing:** Kushan De Silva, Ryan T. Demmer, Daniel Jönsson, Aya Mousa, Helena Teede, Andrew Forbes, Joanne Enticott.

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
