## [Decision Letter · Decision Letter 0]

31 Mar 2022

PONE-D-22-04303Both general- and central- obesity are causally associated with polycystic ovarian syndrome: Findings of a Mendelian randomization studyPLOS ONE

Dear Dr. De Silva,

Thank you for submitting your manuscript to PLOS ONE. After careful consideration, we feel that it has merit but does not fully meet PLOS ONE’s publication criteria as it currently stands. Therefore, we invite you to submit a revised version of the manuscript that addresses the points raised during the review process.

We look forward to receiving your revised manuscript.

Kind regards,

Renato Polimanti, Ph.D.

Academic Editor

PLOS ONE

Journal Requirements:

Reviewers' comments:

Reviewer's Responses to Questions

**Comments to the Author**

1. Is the manuscript technically sound, and do the data support the conclusions?

Reviewer #1: Yes

2. Has the statistical analysis been performed appropriately and rigorously? 

Reviewer #1: Yes

3. Have the authors made all data underlying the findings in their manuscript fully available?

Reviewer #1: Yes

4. Is the manuscript presented in an intelligible fashion and written in standard English?

Reviewer #1: Yes

5. Review Comments to the Author

Reviewer #1: This study asks an interesting question, if anthropometric traits are causal for PCOS. The study is well planned and the statistical analysis is rigurous. As the authors also mention, the most important concern about this study is the low number of cases for PCOS. I suggest meta-analyzing several cohorts (e.g., the UK Biobank and FinnGen) for future studies to increase the sample size.

1. You state that central obesity has a stronger association with PCOS, than BMI. However, you analyze the anthropometric markers, WC, HC, and WHR separately as exposures. Therefore, you have no evidence on ‘central obesity’ as a whole to be causal for PCOS, but you have evidence on separate factors measured for central obesity. Please explain this in the introduction/discussion sections. I suggest running a multivariable MR analysis to understand the complex relationship between all exposures (two different analyses: only markers for central obesity, and markers for central and general obesity) with PCOS as the outcome.

2. To understand the bidirectional relationship between the anthropometric markers and PCOS, it would be interesting to see if PCOS as an exposure is causal for these traits, as we know that the metabolic and hormonal changes in PCOS can lead to weight gain and eventually obesity. Please explain in the discussion the bidirectional relationship between these traits.

3. Please mention in the methods the software you used for the analysis, what I suppose was the TwoSampleMR package in R.

4. I am not sure about the relevance of ‘3.8. Genetic architectures underlying causality’ for this paper. The fact that these anthropometric traits have shared genetic background is already known. The selected IVs were the SNPs that were associated with the exposure dataset, so this part of the manuscript does not help understanding the causality among anthropometric traits and PCOS. In the discussion, ‘A gene set rather similar to those found in the present study was underlying the causal association between BMI and PCOS in a previous MR study as well’ should be explained more thoroughly. What was similar between these genes? What could be their roles?

5. The novel results of this study are WC and HC being causal for PCOS, however, not much is discussed about these results in the discussion section. You should also explain why you think you did not find any association with WHR, another marker for central obesity.

6. PLOS authors have the option to publish the peer review history of their article (what does this mean?). If published, this will include your full peer review and any attached files.

Reviewer #1: **Yes: **Dora Koller

---

## [Author Response · Author response to Decision Letter 0]

15 May 2022

RESPOSNE TO REVIEWERS

We sincerely thank the reviewers for all the comments provided to our initial submission. We have now revised the manuscript accordingly and our point-by-point responses to the comments are given below. Corresponding changes are highlighted in red in the “Revised Manuscript with Track Changes”.

Please note that since our initial submission was made, the outcome cohort has been updated on the IEU OpenGWAS project. The original PCOS cohort named “finn-a-E4_POCS” with a sample size of 96391(219 cases; 96172 controls) has now been updated to “finn-b-E4_POCS” with a larger sample size of 118870 (642 cases; 118228 controls). In order to ensure the reproducibility of our findings based on publicly available data, we therefore re-conducted all analyses, using this larger, updated PCOS cohort. All updated results from these re-analyses are presented in the revised manuscript.

1. This study asks an interesting question, if anthropometric traits are causal for PCOS. The study is well planned and the statistical analysis is rigorous. As the authors also mention, the most important concern about this study is the low number of cases for PCOS. I suggest meta-analyzing several cohorts (e.g., the UK Biobank and FinnGen) for future studies to increase the sample size.

We have acknowledged this in the revised manuscript. Also note that the PCOS cohort used in the updated version is slightly larger than the one used in our original manuscript. As suggested, we have highlighted the need for meta-analyzing several cohorts in future studies, in Discussion section. Specifically, we added the following:

“For future studies, we suggest meta-analyzing several cohorts from sources such as the UK Biobank and FinnGen as a judicious avenue for formidably increasing the sample size.”

2. You state that central obesity has a stronger association with PCOS, than BMI. However, you analyze the anthropometric markers, WC, HC, and WHR separately as exposures. Therefore, you have no evidence on ‘central obesity’ as a whole to be causal for PCOS, but you have evidence on separate factors measured for central obesity. Please explain this in the introduction/discussion sections. I suggest running a multivariable MR analysis to understand the complex relationship between all exposures (two different analyses: only markers for central obesity, and markers for central and general obesity) with PCOS as the outcome.

Anthropometric traits are customarily used as surrogate markers for operationalizing obesity. Previous MR studies assessed total causal effects of central- and general obesity using these anthropometric markers. This has now been explained in Introduction as follows:

“Anthropometric traits are customarily exploited in MR studies as valid and reliable surrogate markers for operationalizing central- and general obesity [30-32]. In these studies, multiple anthropometric markers were incorporated in univariable MR to gain evidence on causal roles of central- and general obesity [30-32].”

Furthermore, as suggested, we conducted multivariable MR analyses to understand the complex relationship between all exposures and findings have been discussed. Findings alluded to the existence of complex causal relationships such as potential causal mediations in the presence of both central- and general obesity and indicated that normal weight-central obesity might have direct causal effects on PCOS. 

Multivariable analyses are described in Methods under “Multivariable MR analyses” section. Findings of multivariable MR are summarized in Table 6 and described in Results under “Multivariable MR findings” section. We have discussed these findings including their implications in Discussion under “Findings of multivariable MR models and their implications” section.

3. To understand the bidirectional relationship between the anthropometric markers and PCOS, it would be interesting to see if PCOS as an exposure is causal for these traits, as we know that the metabolic and hormonal changes in PCOS can lead to weight gain and eventually obesity. Please explain in the discussion the bidirectional relationship between these traits.

This has now been addressed. As described in Methods under “Bidirectional MR analyses”, we conducted MR in the opposite direction, using the same data sources and model specifications and criteria applied to primary univariable MR analyses. However, as described in Results under “Bidirectional MR findings”, sufficient number of functional variants could not be retrieved from existing data. In Discussion, under “Bidirectional associations/Reverse causality”, we have discussed and contextualized findings from previous studies which assessed reverse causality (whether PCOS can cause obesity). All previous bidirectional analyses confirmed that there is no reverse causality. Specifically, we have included the following in Discussion:

“For bidirectional 2SMR analyses, sufficient number of IVs could not be retrieved from existing data sources, but, with the gradual expansion of these open-source GWAS summary data, we envisage that it will be possible to conduct bidirectional analyses more robustly, in the future. Stringent eligibility criteria to select only highly uncorrelated and independent SNPs (LD-R2 = 0.001) may also have contributed to the lack of functional variants produced for assessing reverse causality. However, concordant evidence from previous bidirectional analyses provides reassurance that there is no reverse causality i.e. PCOS would not cause obesity [25, 26].”

4. Please mention in the methods the software you used for the analysis, what I suppose was the TwoSampleMR package in R.

This information is now included in the manuscript. Specifically, in Methods:

“Analyses were conducted in R (version 4.1.2) [44], using “TwoSampleMR” [45], “MRPRESSO” [42], and “RadialMR” [43] packages.”

“This was performed using the multivariable inverse variance weighted method deployed through the ‘mr_mvivw’ function in the ‘MendelianRandomization’ R package [49, 50].”

5. I am not sure about the relevance of ‘3.8’. Genetic architectures underlying causality’ for this paper. The fact that these anthropometric traits have shared genetic background is already known. The selected IVs were the SNPs that were associated with the exposure dataset, so this part of the manuscript does not help understanding the causality among anthropometric traits and PCOS. In the discussion, ‘A gene set rather similar to those found in the present study was underlying the causal association between BMI and PCOS in a previous MR study as well’ should be explained more thoroughly. What was similar between these genes? What could be their roles?

We underscore that MR studies increasingly explore downstream genetic underpinnings including shared genetic basis. As highlighted by previous MR studies, such downstream analyses could help us gain important etiologic insights and clinically valuable signals such as drug targets via functional annotations. This has now been clarified as follows:

“Subsequent to causality testing, MR studies increasingly conduct downstream analyses to explore genetic underpinnings [46, 47]. For instance, previous MR studies showed that functional annotations of highly significant SNPs that act as major drivers of causality in MR models could assist in gaining etiologic insights [46] and identifying putative drug targets [47].”

As shown in S2 Table (Harmonized Data), selected IVs/functional variants associate with both the exposure and the outcome. As summarized in Table 5 and described in Results under “Genetic architectures underlying causality”, we have now explored genetics underlying causality more thoroughly including shared and non-shared SNPs/genes. Furthermore, we have provided evidence on their putative mechanistic roles in S6 Table. The motivation behind this downstream exploration of SNPs/genes underlying causal associations was that it could provide important insights into genetic etiology of PCOS and might provide some directions for future studies aimed at unravelling complex genetic mechanisms of PCOS pathogenesis. This has been discussed in detail in Discussion under “Genetics underlying causality”.

6. The novel results of this study are WC and HC being causal for PCOS, however, not much is discussed about these results in the discussion section. You should also explain why you think you did not find any association with WHR, another marker for central obesity.

We have now discussed the novel findings of this study including WC and HC being causal for PCOS, interpretations and implications of findings on direct causal effects emanating from multivariable MR models, and clinical and public health relevance of these results. Specifically, regarding the importance of causal roles of central obesity demarcated by models comprised of WC and HC, we included following in Discussion:

“In addition, this study offered novel results such as the total causal effects of central obesity on PCOS, potentially direct causal effects of normal weight-central obesity on PCOS, and that complex causal mechanisms in the presence of both general- and central obesity might lead to PCOS. 

Of note, although our analyses indicated possibly direct causal effects of normal weight-central obesity on PCOS, people with central obesity but of normal weight are generally ignored in clinical guidelines [69]. Therefore, women with normal weight-central obesity will require an equally important focus as attributed to general obesity phenotype, in clinical guidelines for PCOS management. To this end, Mediterranean style dietary patterns and physical activity both are evidence-based interventions to reduce central obesity and ensure weight stabilization [70].

Causality of central obesity/adiposity demarcated by WC/HC in the present study corroborate findings from mechanistic studies focused on PCOS etiology. Truncal/visceral obesity promotes insulin resistance which in turn aggravates metabolic and reproductive functions in PCOS such as hyperandrogenism, dyslipidemia, and anovulation [71]. Evidence suggests the formation of a vicious circle by central obesity which involves hyperandrogenemia and truncal adiposity in PCOS [72].

On the whole, findings of this study indicating complex causality of central- and general obesity on PCOS, highlight the need to address the prevention of both forms of obesity at a population level through successful fiscal policies and statutory regulations [73].”

We have also discussed possible reasons for the non-significance of waist-to-hip ratio. Specifically, we included following in Discussion:

“Non-significance of WHR might be owing to its lesser validity and reliability as an anthropometric marker of central obesity, compared to WC/HC, as concluded by previous studies [73, 74]. Lack of adequate statistical power in MR models to uncover causal associations from open-source data might be another contributory factor. This may be verified in future studies using larger cohorts.”

RESPONSE TO OTHER COMMENTS/FORMATTING REQUIREMENTS

Journal Requirements:

The entire manuscript, including all figures and supporting information, has now been formatted according to the stipulated PLOS requirements.

Ethics statement has been included in Methods section.

---

## [Decision Letter · Decision Letter 1]

17 May 2022

Causality of anthropometric markers associated with polycystic ovarian syndrome: Findings of a Mendelian randomization study

PONE-D-22-04303R1

Dear Dr. De Silva,

We’re pleased to inform you that your manuscript has been judged scientifically suitable for publication and will be formally accepted for publication once it meets all outstanding technical requirements.

Kind regards,

Renato Polimanti, Ph.D.

Academic Editor

PLOS ONE

Additional Editor Comments (optional):

Reviewers' comments:

Reviewer's Responses to Questions

**Comments to the Author**

1. If the authors have adequately addressed your comments raised in a previous round of review and you feel that this manuscript is now acceptable for publication, you may indicate that here to bypass the “Comments to the Author” section, enter your conflict of interest statement in the “Confidential to Editor” section, and submit your "Accept" recommendation.

Reviewer #1: All comments have been addressed

2. Is the manuscript technically sound, and do the data support the conclusions?

Reviewer #1: Yes

3. Has the statistical analysis been performed appropriately and rigorously? 

Reviewer #1: Yes

4. Have the authors made all data underlying the findings in their manuscript fully available?

Reviewer #1: Yes

5. Is the manuscript presented in an intelligible fashion and written in standard English?

Reviewer #1: Yes

6. Review Comments to the Author

Reviewer #1: The authors addressed all my comments and added all necessary discussion points that were missing in the first submission. I recommend accepting the manuscript.

7. PLOS authors have the option to publish the peer review history of their article (what does this mean?). If published, this will include your full peer review and any attached files.

Reviewer #1: **Yes: **Dora Koller

---

## [Editor Report · Acceptance letter]

24 May 2022

PONE-D-22-04303R1 

Causality of anthropometric markers associated with polycystic ovarian syndrome: Findings of a Mendelian randomization study 

Dear Dr. De Silva:

I'm pleased to inform you that your manuscript has been deemed suitable for publication in PLOS ONE. Congratulations! Your manuscript is now with our production department. 

Kind regards, 

on behalf of

Dr. Renato Polimanti 

Academic Editor

PLOS ONE